# The force at the implant cannot be assessed by the mallet force–Unless supported by a model

Peter J. Schlieker *, Michael M. Morlock , Gerd Huber

Institute of Biomechanics, TUHH Hamburg University of Technology, Hamburg, Germany

* peter.schlieker@tuhh.de

## Abstract

The implantation of uncemented prostheses requires the application of sufficient forces to achieve a press-fit of the implant in the bone. Excessive forces have to be omitted to limit bone damage. Force measurements along the force transmission path between mallet and implant are frequently used to investigate this trade-off. Placing a load cell at a position of interest (PoI), which might be the implant bone interface or the head taper junction, is technically challenging or even impossible so that nearby positions are chosen. Thus, a certain inertia and stiffness remain between the PoI and the sensor, and consequently the measured dynamic forces differ from those at the PoI. This experimental and numerical study aimed to investigate the amount of force reduction along the transmission path while joining femoral heads to stem tapers. Forces were measured in vitro at the tip of the mallet, directly above the polymer tip of the impactor and below the stem taper. Springs and masses were used to represent the responding tissue of a patient. A semi-empirical numerical model of the force transmission path was developed and validated in order to simulate a larger range of responding tissue properties than experimentally possible and to investigate the influence of different surgical instruments. A distinct attenuation was observed since the peak forces at the impactor reached 35% of the applied peak forces and 21% at the stem taper, respectively. The force curves were replicated with a median root mean square error of 3.8% of the corresponding mallet blow for the impactor and 3.6% for the stem. The force measurement position and the used surgical instruments have a strong influence on the measured forces. Consequently, the exact measurement conditions with regard to sensor positioning and used surgical instruments have to be specified and hence only studies with similar setups should be compared to avoid misestimation of the forces at the PoI. The proposed dynamic numerical model is a useful tool to calculate the impact of the chosen or changed mechanical parameters prior to executing experiments and also to extrapolate the effect of changing the applied forces to the resulting forces at the PoI.

## Introduction

Periprosthetic femoral fractures are one of the major reasons for revision surgery in total hip arthroplasty, involved in 19.7%– 24.6% of revisions [1–3]. Minor intraoperative damages to

**Funding:** The author(s) received no specific funding for this work.

**Competing interests:** I have read the journal's policy and the authors of this manuscript have the following competing interests: MMM is a paid consultant of DePuy Synthes and obtains research support as a Principal Investigator from Ceramtec, DePuy, and Beiersdorf. He obtains speaker's fees from Aesculap, Ceramtec, DePuy, Zimmer, Peter Brehm, Corin, and Mathys and is in the editorial board "Trauma und Berufskrankheit." GH is an associated member of the board of the German Society of Biomechanics.

the femur might be a possible cause that later develop into fractures [4]. These can be introduced by excessive impaction forces, but too little forces do not provide a sufficient press-fit. For the resulting trade-off, crucial force thresholds of the impaction forces are still under debate. The most commonly used source of the implantation force is a mallet, which can generate short and high force impulses when used together with a metal impactor. At the same time, amplitude and number of blows depend on the surgeon [4, 5]. Therefore, automated tools like DePuy Synthes' Kincise™ and IMT's Woodpecker have been developed to reduce the variability of the cavity preparation and, in the case of Kincise™, also of the implantation process.

To improve the force at the Points of Interest (PoI) like the implant-bone interface or the taper junction between head and stem, the force transmission from the mallet or the automated tool to the PoI must be understood. Common methods to measure the impaction force along the transmission path are instrumented mallets [4, 6], instrumented broach handles [7], instrumented impactors [8], instrumented head impactors [9–11], load cells in the stem [5, 12] or beneath the experimental setup. All of the above have in common, that the Position of Measurement (PoM) cannot be the same as the PoI. This leaves a certain inertia and stiffness in between, which affects the measured force in such dynamic processes like a mallet blow. However, the extent of these influences is unknown and so the comparison of forces of different PoM is vague.

The purpose of this experimental and numerical study was to investigate the amount of force lost along the transmission path between the surgical mallet and the PoI. For simplicity reasons, the force transmission during the in vitro assembly of the head taper junction between stem taper and the femoral metal ball head was analyzed, focusing explicitly on the force transmission rather than the taper junction itself.

## Materials and methods

Mallet impaction is commonly used in orthopedics. For simplicity reasons, the exemplary setup to investigate the force transmission path was the assembly of metal ball heads to stem tapers.

### In vitro experiment

Heads (CoCr, ⌀ 28 mm, 12/14 taper, DePuy Synthes, Raynham, MA; n = 8) were placed loosely on stem taper replicas (Ti-6Al-4V, 12/14 tapers, 48 mm long, DePuy Synthes; n = 8) and assembly was obtained by a single blow of a mallet (0.87 kg; Fig 1A) on top of an impactor (Emphasys head impactor, DePuy Synthes) which was guided vertically to ensure that the axes of the impactor and the taper remained aligned. Before the consecutive assembly, the stem taper replicas were connected to a high mass platform (167 kg) by replaceable elastic springs to simulate different tissue impedances (Fig 1B). A linear guidance constrained horizontal displacements, rotation and inclinations of the stem tapers. The combined weight of all components below the stem taper was 0.52 kg and could be increased representing changes in the weight of the femur or the implant's stem resulting in additional impedances of the responding tissue. After each test, the junctions were disassembled and both parts were cleaned with isopropyl alcohol (≥ 99.8%).

A total of 20 consecutive assemblies with randomly reassigned heads were performed for each stem taper of which the 1st assembly served as preconditioning. During the following assemblies, different stiffnesses and additional masses were combined below the stem taper. The 2nd, 5th, 10th and 15th assembly were conducted with identical parameters (4.0 N/mm, no additional mass) denoted as reference measurements in order to ensure that no bias was

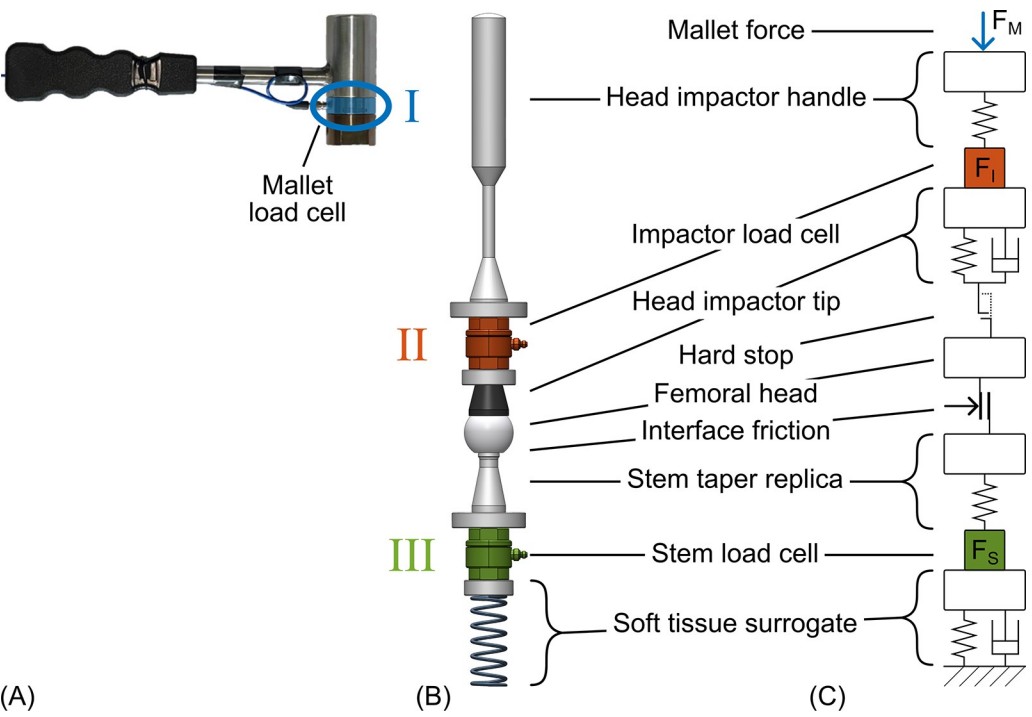

**Fig 1. Experimental and numerical setup.** (A) Mallet with load cell. (B) Simplified experimental setup with load cells in the impactor and below the stem taper replica. The linear guidance to constrain all movements apart of the one-dimensional displacement is not shown. (C) Structure of the semi-empirical numerical model.

introduced by continuously reusing the components. The stiffness of 4.0 N/mm was chosen as reference because it was almost identical to the stiffness determined in vivo by Doyle et al. (4.1 N/mm) [13]. A part of the selection of tissue impedances and different assembly procedures were performed without being related to this research question and are mentioned here only for completeness. This leads to n = 64 measurements evaluated of which 32 were reference measurements, 24 were tested with varying stiffnesses and 8 had additional weight mounted (Table 1).

Forces were measured I) at the striking surface of the instrumented mallet (9041A, Kistler, Winterthur, CH; Fig 1A), II) between the head impactor handle and its corresponding tip

**Table 1. Variations of the soft tissue surrogate for the different consecutive assemblies.** The stiffness of 4.0 N/mm of the reference groups was chosen based on results from Doyle et al. measured in full body cadavers (4.1 N/mm [13]). The other stiffnesses were selected in a range from 15% to 125% of the reference stiffness in order to consider a broader spectrum of tissue properties.

| Assembly | Mass [kg] | Stiffness [N/mm] |
|---|---|---|
| 1 (preconditioning) | 0.52 | 4.0 |
| 2, 5, 10, 15 (reference) | 0.52 | 4.0 |
| 3 | 0.52 | 1.9 |
| 6 | 0.52 | 5.0 |
| 8 (additional weight) | 0.89 | 4.0 |
| 13 | 0.52 | 0.6 |
| 4, 7, 9, 11, 12, 14 | Additional measurements and different assembly procedures not related to this research question | |

Each set of assembly parameters was used for all eight stem tapers.

(Emphasys femoral head impactor tip, DePuy Synthes; polyether ether ketone), and III) at the stem below the taper (both 9333A, Kistler; Fig 1B). Data acquisition was performed at 800 kHz (NI-9775, National Instruments, Austin, TX). Noise and oscillations caused by the stiffness of the load cells were removed from the force signals by filtering with 4$^{th}$ order zero-phase Butterworth low-pass filters. The cut-off frequencies were adapted to the PoM since each load cell had a unique combination of sensor stiffness and attached weights (I: 20 kHz, II: 10 kHz, III: 15 kHz).

## Numerical model

For the uniaxial semi-empirical numerical model, the major components of the force transmission path were simplified with ideal physical components ([14], Fig 1C; Simscape, Simulink 2022B, MathWorks, Natick, MA). Additionally, the loose contact between the impactor tip and the mass of the head as well as the indentation of the metal head into the softer impactor tip were represented by a one-sided hard stop with linearly increasing stiffness within a transition region. A friction element was used to simulate the clamping at the head taper junction by considering the friction coefficients and the estimated normal force between head and stem taper. The normal force is proportional to the contact area of the taper junction [15] hence a polynomial of degree two related to the insertion depth of the stem taper into the head taper was used but without a linear term.

From the measured mallet force solely the first peak was used as force input at the striking surface of the simulated impactor. Since the experimental setup was vertical, the gravitational force of all mass components was implemented in axial direction. In addition, the initial compression of all spring elements was set according to each gravitational load to start in a steady state without movements before the mallet impact. The simulation duration was 0.8 ms and the fundamental sampling frequency was fixed to 800 kHz, according to the frequency of the data acquisition during the experimental part.

The model included ten parameters that could not be determined exactly (impactor stiffness, impactor tip stiffness and damping, transition region of the hard stop between the impactor tip and the head, static and kinetic friction coefficients at the taper junction, constant term of the normal force at the taper junction, the displacement-dependent factor of the normal force at the taper junction, stem stiffness, and damping of the responding tissue). Therefore, a sensitivity analysis was performed on one preliminary experimental dataset to detect parameters with little influence by using Monte Carlo simulations and correlation analysis [16] to decrease the degrees of freedom for the following parameter optimization.

The transition region of the hard stop between the impactor tip and the head, the constant term of the normal force at the taper junction, and the damping of the responding tissue were rated to be less influential and were therefore assigned broadly estimated values of 50 μm, 5 N, and 10% of the corresponding spring stiffness respectively. Afterwards, the remaining seven parameters–impactor stiffness, impactor tip stiffness and damping, static and kinetic friction coefficients, the displacement-dependent factor of the normal force at the taper junction, and stem stiffness–were determined by optimizing the sum squared error of the forces at the impactor and below the stem taper for all datasets of the repeatedly assembled reference groups.

The goodness of fit between simulated and measured data was evaluated by calculating the root mean square error (RMSE) for the attenuated forces at the impactor and the stem taper. Therefore, for each PoI the values from the zero crossing before the first to the zero crossing after the second peak were considered. This includes two positive half-waves and the negative in between.

The model was used for two extrapolations. First, the mass of the impactor was increased gradually in five steps between 0.1 kg and 2 kg. For each mass 26 different stiffnesses between 100 N/mm and 1 x 10$^6$ N/mm were used for the impactor stiffness resulting in 130 simulations for the investigation of the influence of the mechanical properties of the impactor on the force transmission. Second, for the parameters of the analogous model of the responding tissue, eight masses between 0.1 kg and 20 kg were combined with eight stiffnesses in the range 0.1 N/mm to 50 N/mm resulting in 64 simulations. The relatively large ranges were chosen to give a more general overview of the dependencies on the varied parameters, even though the realistic values may lie in a smaller range.

## Statistical analysis

Statistical analysis was performed with a type I error level of α = 0.05 (SPSS 26.0, IBM, Armonk, NY). Normality and homogeneity of variance were checked with Shapiro-Wilk and Levene's test respectively. When requirements were fulfilled, one-way analysis of variance with Bonferroni correction was used to analyze differences between means, otherwise Kruskal-Wallis test together with Mann-Whitney U test as post hoc test were performed. When only two groups were compared, an independent t-test or Mann-Whitney U test was used. Results of parametric tests were presented as bar charts with mean ± standard deviation. Non-parametric results were visualized as box plots with whiskers up to 1.5 interquartile ranges (IQR) in length. Values outside this range were highlighted as outliers but not excluded from further evaluations.

## Results

### Experiment

The exerted mallet blows resulted in the desired short and high impulses (9247 N ± 1513 N, 0.140 ms ± 0.008 ms). All measured forces were normalized to the peak force of the corresponding mallet blow to account for the unavoidable differences between the manually applied blows. The resulting normalized force peaks represent the attenuation of the forces from mallet to impactor and from mallet to stem. Force signals of the reference groups showed reproducible curves for each PoM after filtering and normalization (Fig 2) and were not affected by the number of consecutive assemblies (impactor: p = 0.135, stem: p = 0.396). The force signals of the impactor resembled a slightly damped sinus with some offset for the first millisecond (first peak 34.8% ± 1.3%, period 0.27 ms (IQR: 0.26 ms to 0.28 ms)). The forces of the stem also behaved sinusoidal with an increasing offset. Thus, the highest force was reached at the second peak (second peak 21.2% ± 3.4%, period 0.15 ms (IQR: 0.14 ms to 0.16 ms)). The attenuation of the peak force from mallet to impactor to stem was significant (both PoM: p < 0.001). The time delay of the peaks between the different PoM increased with distance to the mallet (mallet-impactor: 0.0694 ms ± 0.0024 ms, mallet-stem: 0.4624 ms ± 0.0250 ms, p < 0.001). The significance in the latter case was independent of whether the first or the second local maximum of the force of the stem was considered.

The stiffness of the soft tissue surrogate did not influence the force peaks (Fig 3, impactor: p = 0.187, stem: p = 0.534)–at least not for the examined range–and did also not influence the time delays (Fig 4, impactor: p = 0.102, stem: p = 0.421) at either PoM. Increasing the mass of the stem piece resulted in an upwards shift of the force curves of the stem with changed oscillatory behavior after approximately 0.6 ms (Fig 5). Accordingly, the force of the stem increased significantly (Fig 6, p = 0.001) while all other evaluation variables remained unchanged (impactor force: p = 0.654, impactor delay: p = 0.517, stem delay: p = 0.561).

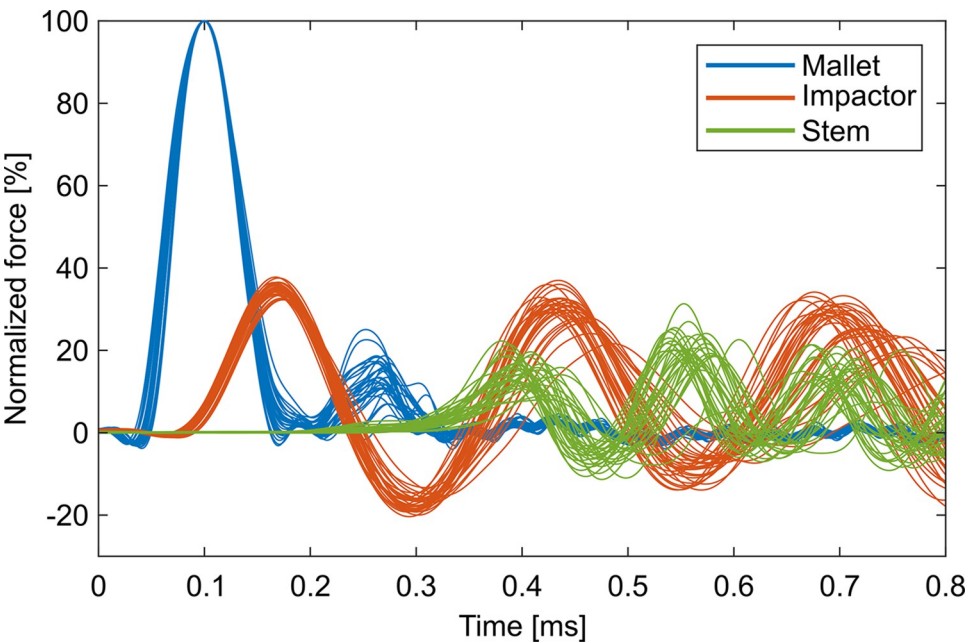

**Fig 2. Normalized force measurements of the mallet, impactor and stem.** The 32 reference groups exhibited rather similar characteristics for each PoM including the attenuation from mallet to impactor and stem.

## Numerical model

During the parameter estimation process, three datasets were excluded from the optimization because they showed distinctly different oscillatory behavior. The remaining 29 datasets

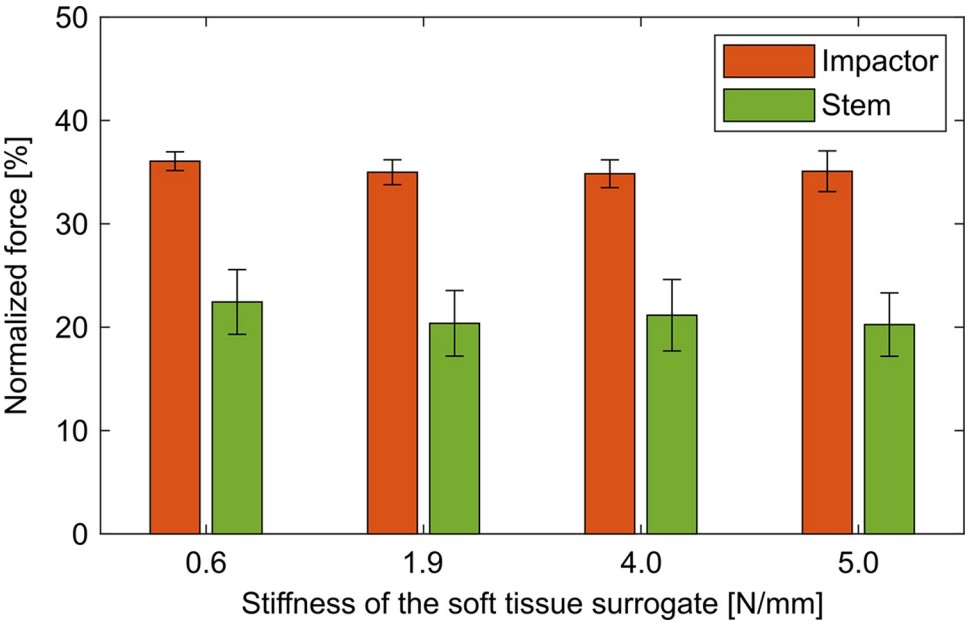

**Fig 3. Normalized force in dependence of the stiffness of the soft tissue surrogate.** The stiffness of the soft tissue surrogate did not affect the attenuation of the force for the values considered in this study–neither for the impactor nor for the stem.

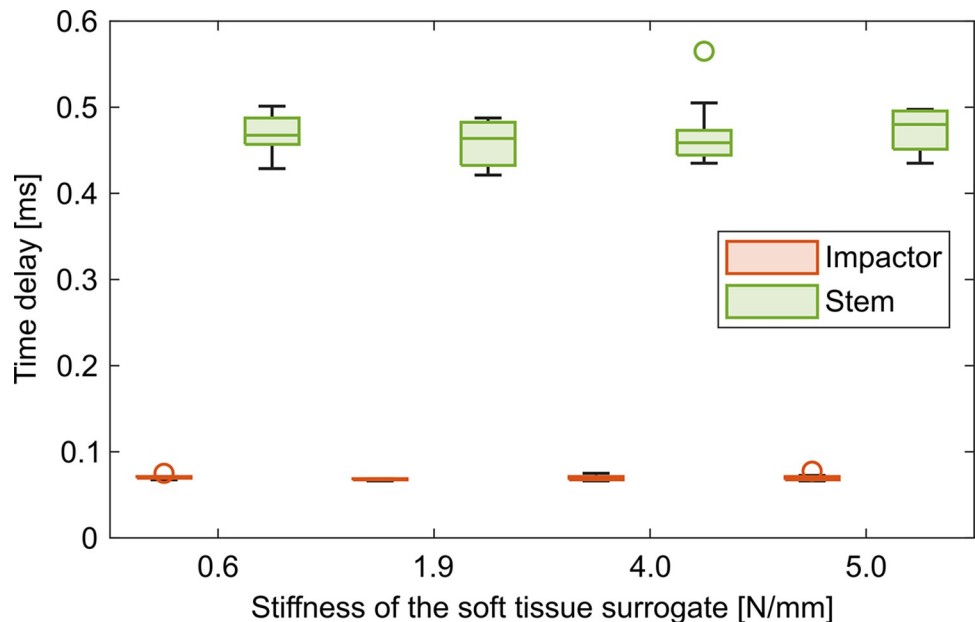

**Fig 4. Time delay of force peaks in dependence of the stiffness of the soft tissue surrogate.** The stiffness did not affect the time delay for the values considered in this study–neither for the force of the impactor nor for the force of the stem.

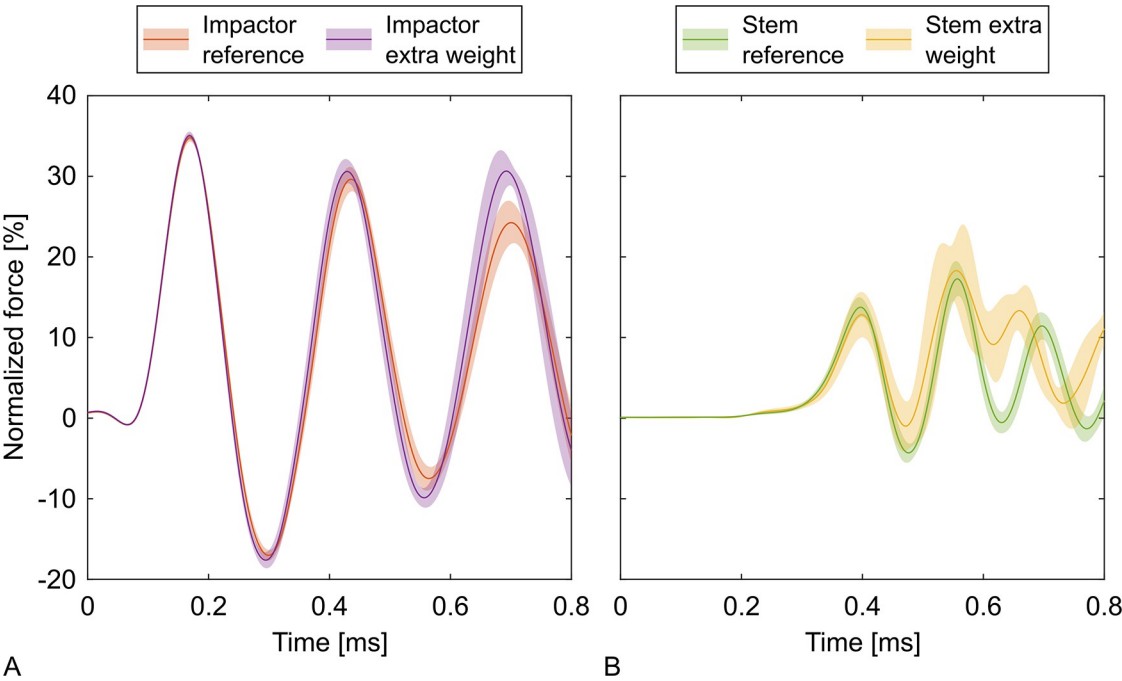

**Fig 5. Normalized forces for the additional weight below the stem.** The curves represent the mean normalized forces with 95% confidence intervals for the impactor and the stem for the reference groups and the assemblies with additional weight below the stem (+0.37 kg, +71%). The additional weight had little effect on the force of the impactor. The force of the stem was shifted upwards due to the additional weight and showed a different oscillatory behavior after approximately 0.6 ms.

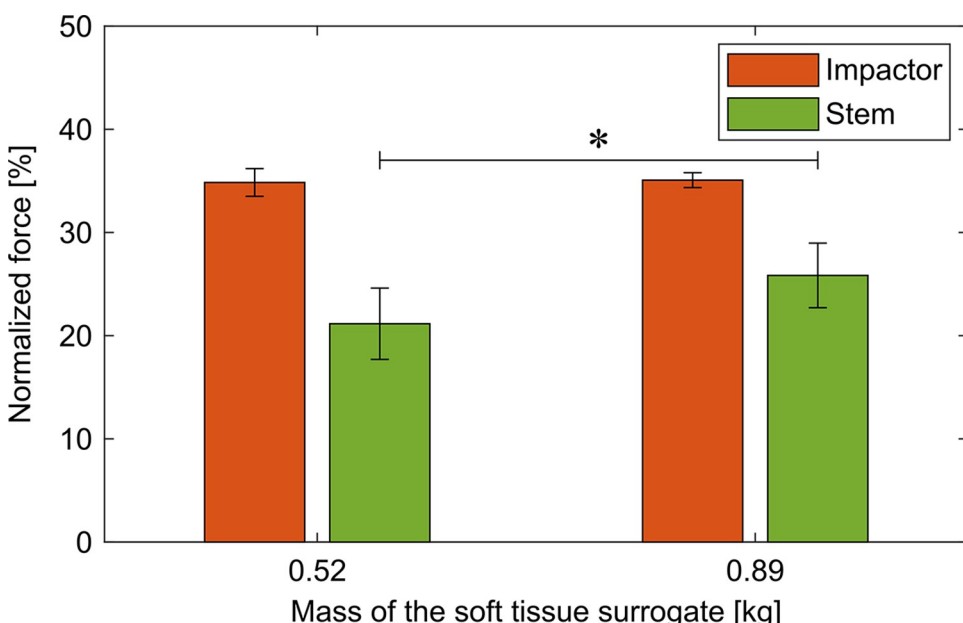

**Fig 6. Normalized peak forces for the additional weight below the stem.** Increasing the weight at the stem resulted in higher forces at the stem. In contrast, the forces at the impactor were not affected by the changed weight.

formed the baseline for the model and the identified parameters were used to simulate all performed experiments (Table 2).

For the reference groups the numerical model obtained a RMSE of 3.81% (IQR: 3.35% to 4.77%) for the impactor and 3.60% (IQR: 2.55% to 4.50%) for the stem. The peaks of the attenuated impactor forces were consistently underestimated during the first two oscillations (Fig 7). The time offset of the peaks was 3.75 μs (IQR: 2.50 μs to 6.25 μs; equivalent to 3 sampling periods (IQR: 2 to 5 sampling periods)) for the impactor and -2.50 μs (IQR: -8.75 μs to 10.00 μs; equivalent to -2 sampling periods (IQR: -7 to 8 sampling periods)) for the stem.

The model that was optimized for the reference measurements was also able to predict the results of the measurements that were generated with different soft tissue surrogates. The RMSEs were similar (impactor: p = 0.504, stem: p = 0.148) and neither the force offsets

**Table 2. Parameters of the numerical model including the results of the parameter estimation.**

| Object | Mass [kg] | Stiffness [N/mm] | Damping [Ns/mm] | Miscellaneous |
|---|---|---|---|---|
| Impactor | 0.508 | $85 \times 10^3$ | | |
| Load cells | 0.137 | $2.3 \times 10^6$ | | |
| Impactor tip | 0.084 | $13 \times 10^3$ | 240 | |
| Transition between impactor tip and head | | | | $50 \times 10^{-6}$ m |
| Femoral head | 0.062 | | | |
| Friction inside head taper junction | | | | $\mu_{kinetic} = 0.49$<br>$\mu_{static} = 0.79$ |
| Normal force of head taper junction | | | | 5.0 N<br>$140 \times 10^9$ N/m$^2$ |
| Stem taper replica | 0.043 | $130 \times 10^3$ | | |
| Analogous model of the responding tissue | 0.52* and 0.89* | 0.6 to 5.0 | 10% of stiffness | |

* including load cell

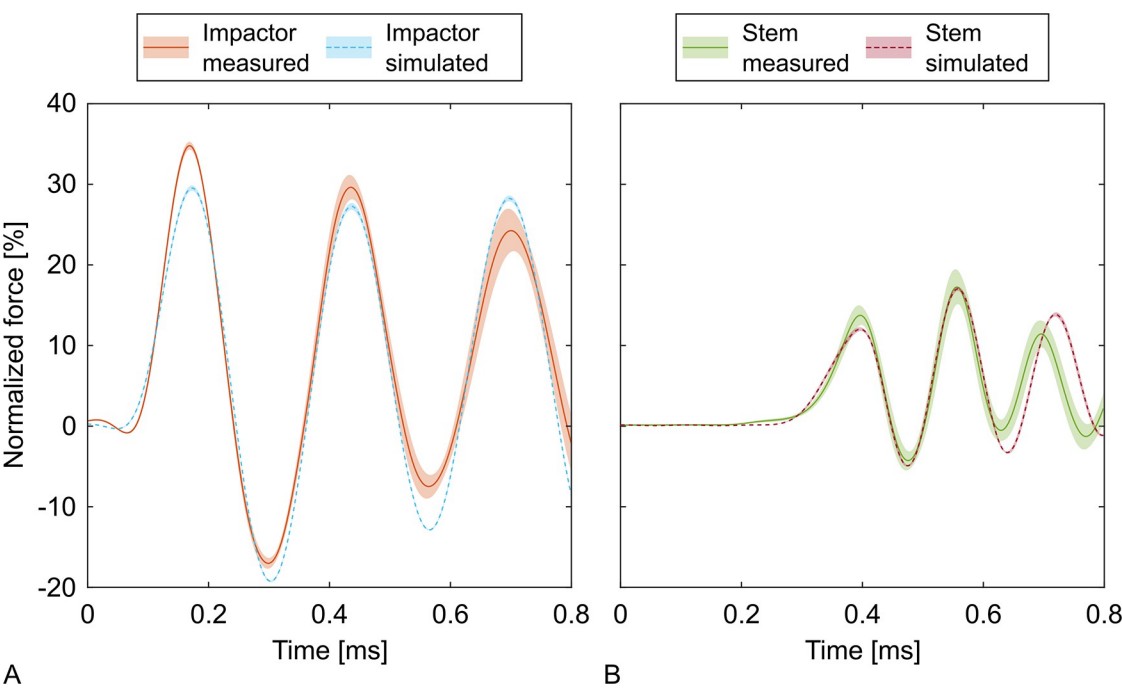

**Fig 7. Measured and simulated forces of the impactor and stem.** The curves represent the normalized forces of the reference groups averaged for each point in time together with their 95% confidence intervals. The confidence intervals of the simulations are smaller than the line width in some areas.

(impactor: p = 0.097, stem: p = 0.586) nor the time offsets (impactor: p = 0.173, stem: p = 0.218) were affected by the changes of the surrogate.

Simulation results for the large selection of mass and stiffness of the impactor showed that the force of the impactor was greater the lighter the impactor was (Fig 8A). In addition, heavier impactors resulted in flatter curves and the peaks were reached at higher impactor stiffnesses. In the area of the estimated stiffness and the actual weight of the impactor, the impactor force increased logarithmically with the stiffness and decreased with increasing weight. The force of the stem at the actual weight from the experiments was barely affected by the variation of the stiffness of the impactor (Fig 8B). Lighter impactors showed a strong dependence on the stiffness with maximum forces of the stem becoming higher and getting closer to the estimated stiffness at lower weights of the impactor.

The force of the impactor was not affected by the different parameters of the analogous model of the responding tissue and was constantly 29.2%. Higher masses resulted in higher forces below the stem (Fig 9). When high masses were used, the variation of the stiffness had no effect on the force below the stem. Stiffnesses above the spring stiffnesses used during the experiments increased the forces of the stem for the lighter analogous models of the responding tissue.

## Discussion

The derived force attenuation in the transmission path from mallet to stem exemplarily focused on the assembly of the head taper junction, but similar mechanisms are likely elsewhere in the implantation process (e.g. reaming and insertion of the femoral stem), since the stiffnesses and masses are of a similar order of magnitude.

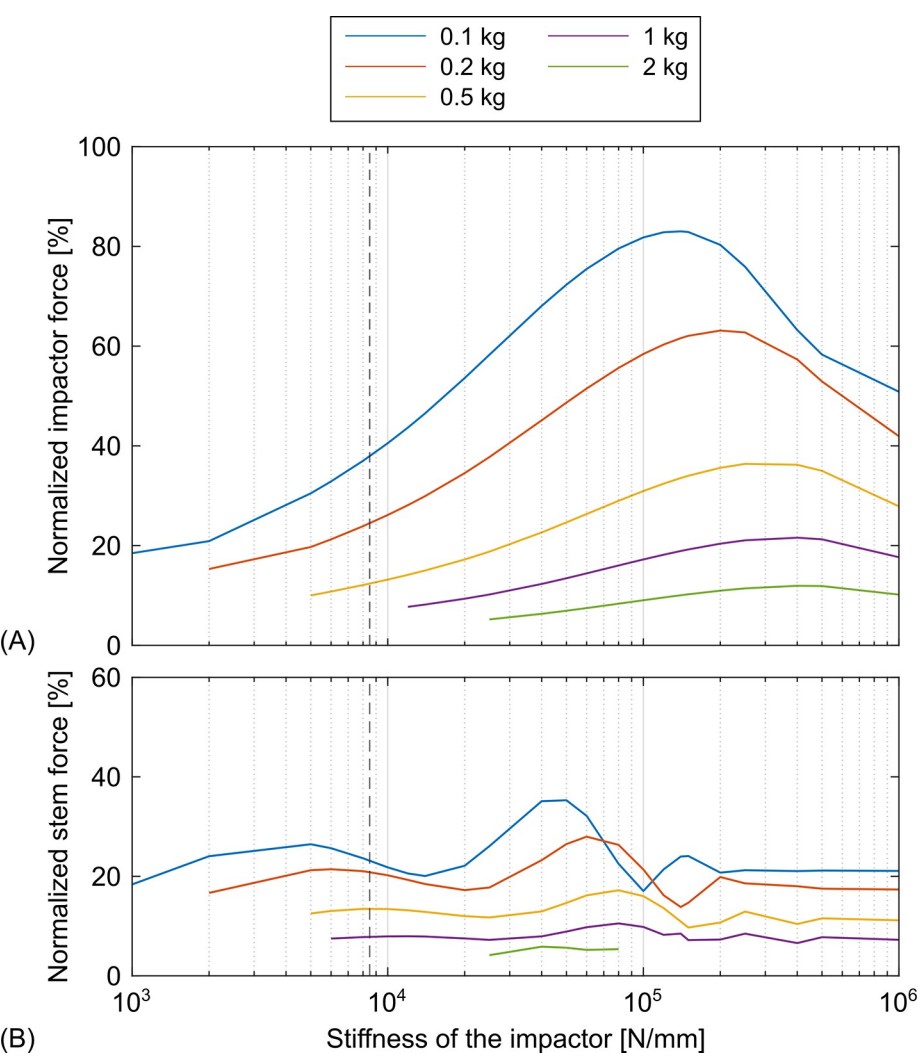

**Fig 8. Parameter variation of the impactor.** Assemblies with a large selection of masses and stiffnesses of the impactor were simulated. Low stiffnesses and high masses of the impactor did not result in oscillations with determinable peaks within the simulation time and were therefore excluded, resulting in curves of different lengths. The stiffness from the parameter estimation was marked with a dashed vertical line. The green line corresponds to the real weight of the impactor. (A) normalized force of the impactor and (B) normalized force of the stem.

The cut-off frequencies of the mallet and impactor filters were about five times higher than the most dominant frequency of the corresponding signal. This ratio was sufficient to filter the desired frequencies without substantially distorting the signals.

The introduced semi-empirical numerical model was capable to reproduce the measured force curve in the time domain, even though minor variations in frequency resulted in relatively large increases of the RMSEs. The remaining underestimation of the peaks is most likely due to the simplifications caused by the usage of basic mechanical components. Since the force can be calculated at any position in the model, it can be used to estimate forces at PoIs which are not accessible with sensors. Beside this, the validated and parametrized model could also simulate most of the observed behavior of the experiments using different configurations for the analogous model of the responding tissue–without repeating the parameter optimization.

Along the transmission path the impaction force was unexpectedly attenuated, which has to be considered if force measurements from different PoMs are compared. Each PoM should be

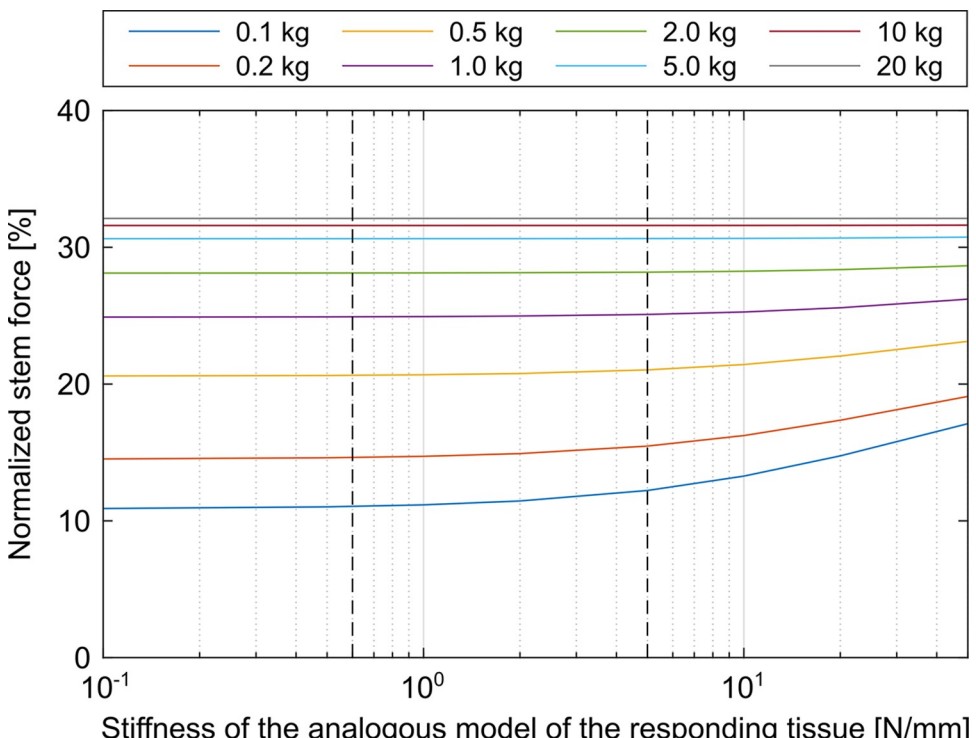

**Fig 9. Parameter variation of the analogous model of the responding tissue.** Assemblies with a large selection of masses and stiffnesses of the responding tissue were simulated. The range of stiffnesses used in the experiments was marked with two dashed vertical lines. The yellow and purple line correspond to the approximate weights considered in the experiments.

as close and as rigidly connected to the PoI as possible to receive accurate forces for the PoI. Even the results of different studies with similar PoMs may strongly differ since different surgical instruments–that differ in mass and stiffness–are likely to be used. All these points can be addressed by using a model like the one presented in this paper to calculate the forces at the PoIs while considering the complete experimental setup.

Wendler et al. reported a less severe attenuation at the implant neck that was 84.4% ± 13.5% of the introduced mallet force during head taper assembly on human cadavers [12]. They attributed the attenuation to their bearing of the handheld impactor or a misalignment between the impact direction and the neck axis [12]. However, based on the simulations performed, it could be shown that friction is not the decisive factor for the attenuation. Scholl et al. already suggested that components in the transmission path between force application (mallet) and the PoM reduce the peak force at the PoM [4]. The presented model however helped to relate the observed effects to the inertia and stiffness of the components–leading to the severe attenuation to less than 40% of the introduced impaction force in this study.

The following limitations have to be mentioned. In clinical practice, heads and stems are not reused to avoid potential negative influences [17]. However, in this particular study the junction mechanics were not in the scope and repeated measurements of the reference groups showed that reuse had no effect on the measured forces within the force transmission path. The force impulse and the impactor axis might not always have been in line, due to the manual excitation by a mallet, potentially causing minor differences between measured mallet force and acting force in axial direction of the impactor–in vitro and in clinics. Furthermore, it is possible that variations within the measurements occurred due to screws tightened to different degrees, despite regular retightening of all screws.

Alike to Krull et al., the results suggest that the stiffness of the surrounding tissue has a minor influence on the implantation forces [9, 18]. The stiffness used for the reference groups was taken from Doyle et al., but since the cadavers they used were rather light (mean weight: 60 kg, range: 50 kg to 65 kg) [13], heavier patients with more surrounding soft tissue may not have been covered. The stiffnesses of the additionally used springs were in the same order of magnitude, but at this point it is not possible to assess whether this represents the physiological range. Therefore, a wider range of parameters for the responding tissue was covered in the simulation. The occurring increase of the force below the stem for high stiffnesses in combination with low masses of the responding tissue can be explained with the natural frequency of the tissue. The impulse of the impactor has a frequency of approximately 3.5 kHz which is clearly overcritical for the tissue parameters from Doyle et al. with a natural frequency of almost 3 Hz [13]. With increasing stiffness, the natural frequency of the tissue increases and the excitation gets closer to the resonance case. A wide range of patient-specific tissue parameters, positioning and fixation of the patient during the surgical procedure and new surgical methods [19] could further affect the mechanical situation and should therefore be investigated in further projects.

## Conclusion

The result of any force measurement in the transmission path of a surgical impaction highly depends on the chosen PoM and the included impaction instruments. It is mandatory to report that information in order to assess whether a comparison between studies is possible. The model presented can be used to investigate influences of design changes of surgical instruments or implants on the force transmission to narrow down the number of experimental tests to be performed prior clinical introduction. The model could also allow a more accurate determination of the forces at other PoIs, for example at the implant-bone interface.

## Supporting information

**S1 File. Contains all data required to recreate the figures from the manuscript.**
(ZIP)

## Acknowledgments

The heads, tapers and surgical instruments were kindly provided by DePuy Synthes. The authors greatly appreciate the support of Kay Sellenschloh and Matthias Vollmer during the experiments.

## Author Contributions

**Conceptualization:** Peter J. Schlieker.

**Data curation:** Peter J. Schlieker.

**Formal analysis:** Peter J. Schlieker.

**Investigation:** Peter J. Schlieker.

**Methodology:** Peter J. Schlieker.

**Resources:** Michael M. Morlock, Gerd Huber.

**Software:** Peter J. Schlieker.

**Supervision:** Michael M. Morlock, Gerd Huber.

**Writing – original draft:** Peter J. Schlieker.

**Writing – review & editing:** Michael M. Morlock, Gerd Huber.

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
