## [Decision Letter · Decision Letter 0]

21 Feb 2024

PONE-D-23-37167The force at the implant cannot be assessed by the mallet force – unless supported by a modelPLOS ONE

Dear Dr. Schlieker,

Thank you for submitting your manuscript to PLOS ONE. After careful consideration, we feel that it has merit but does not fully meet PLOS ONE’s publication criteria as it currently stands. Therefore, we invite you to submit a revised version of the manuscript that addresses the points raised during the review process.

We look forward to receiving your revised manuscript.

Kind regards,

John Leicester Williams, Ph.D.

Academic Editor

PLOS ONE

Journal Requirements:

Additional Editor Comments:

The reviews have been favorable overall. The reviewers have commented on a number of points that they felt were unclear or that need further explanation. Please address these in your revision and rebuttal statement.

Reviewers' comments:

Reviewer's Responses to Questions

**Comments to the Author**

1. Is the manuscript technically sound, and do the data support the conclusions?

Reviewer #1: Yes

Reviewer #2: Partly

2. Has the statistical analysis been performed appropriately and rigorously? 

Reviewer #1: Yes

Reviewer #2: Yes

3. Have the authors made all data underlying the findings in their manuscript fully available?

Reviewer #1: Yes

Reviewer #2: Yes

4. Is the manuscript presented in an intelligible fashion and written in standard English?

Reviewer #1: Yes

Reviewer #2: Yes

5. Review Comments to the Author

Reviewer #1: The study is interesting, some details need to be further provide to further improve the overall quality.

Please indicate the rationale on the selection of the Stiffness in Table 1.

Line 109 (please describe the model parameters with more details, please better clarify the sensitivity analysis (lines 109-111).

The results (Figure 2) show high reproducibility on the first peaks, less on the second one, do the author have an explanation Why stem curves have lower reproducibility?

Figure 3 and 4 please add the sentence ‘for the values considered in this study.’

figure 7 illustrates the difference between the measured and the simulated impactor force. looking to the first peak some difference could be identified. How the author could claim it is validated?

Why the results of the simulator are understimated?

the authors used metal head, do the author consider this material as relevant or irrelevant for their results, in other words, the use of ceramic or Oxinium head could have altered the results or the one find are also valid for other type of materials?

Reviewer #2: Indroduction

1. The authors explain the relationship between the forces required to achieve a sufficient press fit and excessive forces that lead to periprosthetic femoral fractures. In the experiments, however, the junction between the femoral head and the stem was investigated. These are two different situations. Please explain why you describe risks and instruments for the insertion of the stem or cavity preparation, but examine the taper junction in your study. (line 37-54)

Materials Methods

1. The measurement signals were digitally filtered after the measurement to suppress noise. The cut-off frequencies are in the range of the frequencies contained in the force signals (period duration of 0.1 ms leads to frequencies of >10 kHz). Has it been investigated whether the cut-off frequencies influence the shape of the measured signals? This may have an influence on the goodness of fit between measurement and simulation.

Discussion

1. Is it necessary to measure the force in the PoI (head taper junction)? Previous investigations have shown a linear relationship between forces on the stem and turn-off moments or pull-off forces.

Conclusion

1. You mention that, based on your findings, it is necessary to provide all the information about the instruments used in experimental studies. In your opinion, is it possible to convert study results via the model and thus make them comparable?

2. The implant-bone interface was neither mentioned in the results nor in the discussion. Therefore, it cannot be stated in the conclusion that the forces at the implant-bone interface can be determined more accurately. The corresponding interface has not been modelled in the mechanical model. In my opinion, both the model and the experimental setup would have to be extended in order to be able to make such a statement.

6. PLOS authors have the option to publish the peer review history of their article (what does this mean?). If published, this will include your full peer review and any attached files.

Reviewer #1: **Yes: **Bernardo Innocenti

Reviewer #2: No

---

## [Author Response · Author response to Decision Letter 0]

4 Apr 2024

RESPONSES TO THE REVIEWER COMMENTS

The authors would like to thank the editor and reviewers for their valuable feedback and helpful comments. All comments were answered to the best of our ability, so hopefully all doubts have been clarified.

ACADEMIC EDITOR:

COMMENT 1

RESPONSE 1

The requirements have been checked and the manuscript complies with the guidelines.

COMMENT 2

Please note that PLOS ONE has specific guidelines on code sharing for submissions in which author-generated code underpins the findings in the manuscript. In these cases, all author-generated code must be made available without restrictions upon publication of the work. Please review our guidelines at https://journals.plos.org/plosone/s/materials-and-software-sharing#loc-sharing-code and ensure that your code is shared in a way that follows best practice and facilitates reproducibility and reuse.

RESPONSE 2

The complete model including the results of the parameter optimization has been submitted to a repository, is cited in line 100 and is now listed in the references with its own DOI. The Data Availability Statement was modified accordingly. As soon as the DOI for this manuscript is available, it will be published and made freely available online.

COMMENT 3

Please include captions for your Supporting Information files at the end of your manuscript, and update any in-text citations to match accordingly. Please see our Supporting Information guidelines for more information: http://journals.plos.org/plosone/s/supporting-information.

RESPONSE 3

The following caption was added for the supporting information file “S1 Supporting material. Contains all data required to recreate the figures from the manuscript.” and the file was renamed accordingly.

 

COMMENT 4

RESPONSE 4

The reference list was checked and the following changes were made:

The references 1 (NJR 2023) and 2 (AOANJRR 2022) were updated as they were accessed again on 04 April 2024.

Reference 7 (Konow et al. 2023) was updated with a DOI as it was accepted and published in the meantime.

The reference for instrumented impactors (line 48) was changed to “Glismann K, Konow T, Huber G, Lampe F, Ondruschka B, Morlock MM. Small design changes affecting primary stability in fully coated tapered wedge stems. PLOS ONE. 2024:Forthcoming” which was therefore added to the reference list as reference 8.

According to comment 3 from reviewer 1, reference 16 “The MathWorks, Inc. Simulink Design Optimization Reference. 2022 Available from: https://de.mathworks.com/help/releases/R2022b/pdf_doc/ sldo/sldo_ref.pdf (date last accessed 04 April 2024).” was added to the reference list.

COMMENT 5

The reviews have been favorable overall. The reviewers have commented on a number of points that they felt were unclear or that need further explanation. Please address these in your revision and rebuttal statement.

RESPONSE 5

The authors sincerely appreciate your and the reviewer’s feedback and have thoroughly processed all comments.

 

REVIEWER #1:

COMMENT 1

The study is interesting, some details need to be further provide to further improve the overall quality.

RESPONSE 1

Thank you for your positive feedback in order to further improve the quality of the manuscript. We appreciate all of your comments and will consider them in detail.

COMMENT 2 (MATERIALS & METHODS)

Please indicate the rationale on the selection of the Stiffness in Table 1.

RESPONSE 2

We are grateful for your hint that this point was not clear enough. We have added the following sentence to the caption of Table 1 (line 84) to explain the selection of stiffnesses:

The stiffness of 4.0 N/mm of the reference groups was chosen based on results from Doyle et al. measured in full body cadavers (4.1 N/mm [13]). The other stiffnesses were selected in a range from 15% to 125% of the reference stiffness in order to consider a broader spectrum of tissue properties.

Comment 3 (MATERIALS & METHODS)

Line 109 (please describe the model parameters with more details, please better clarify the sensitivity analysis (lines 109-111). 

RESPONSE 3

Thank you for pointing that out. We have added the following term in parentheses in line 113 for a better understanding of the model parameters.

(impactor stiffness, impactor tip stiffness and damping, transition region of the hard stop between the impactor tip and the head, static and kinetic friction coefficients at the taper junction, constant term of the normal force at the taper junction, the displacement-dependent factor of the normal force at the taper junction, stem stiffness, and damping of the responding tissue)

For a better understanding of the sensitivity analysis “by using Monte Carlo simulations and correlation analysis [16]” was added in line 117.

 

COMMENT 4 (RESULTS)

The results (Figure 2) show high reproducibility on the first peaks, less on the second one, do the author have an explanation Why stem curves have lower reproducibility?

RESPONSE 4

Simulations use to become less accurate with longer simulation time and with increasing distance (mallet to impactor curve and impactor to stem) from the input signal. To avoid this, more complex models would be needed, but this would be associated with a higher number of uncertain parameters (e.g. heads were manually placed on the stem tapers, differences in initial assembly between the samples could not be avoided completely which might lead to a larger variation between samples). Thus, the concept of modeling – simplification – comes along with less reproducibility of less important outputs (the second peak in this case). We will further elaborate this issue in response 7.

COMMENT 5 (RESULTS)

Figure 3 and 4 please add the sentence ‘for the values considered in this study.’

RESPONSE 5

Done!

COMMENT 6 (RESULTS)

figure 7 illustrates the difference between the measured and the simulated impactor force. looking to the first peak some difference could be identified. How the author could claim it is validated?

RESPONSE 6

We greatly appreciate the critical perspective you offer. The model is capable of rebuilding the complex shape of the measured signal quite well. Only the amplitudes close to the peaks are not a perfect fit but the domain is correct. The focus was placed on the frequencies and phase shift which have been successfully reduced to almost zero. Since the model is a simplification of the real situation, a certain (in our opinion acceptable) difference will always remain.

COMMENT 7 (RESULTS)

Why the results of the simulator are understimated?

RESPONSE 7

We are thankful for your question. We assume that the underestimation of the forces by the model is due to the fact that the model was built from basic mechanical components. This leads to simplifications of the complex and highly dynamic impaction scenario including the assumption that all components behave linearly.

As clarification the following sentence was added into the discussion section “The remaining underestimation of the peaks is most likely due to the simplifications caused by the usage of basic me-chanical components.” (line 241).

 

COMMENT 8

the authors used metal head, do the author consider this material as relevant or irrelevant for their results, in other words, the use of ceramic or Oxinium head could have altered the results or the one find are also valid for other type of materials?

RESPONSE 8

If the different weights of these heads and possibly also changes in friction between the head and the metal taper are considered in the model, the results can also be transferred to other materials.

We thank you for your detailed review of our manuscript.

REVIEWER #2

COMMENT 1 (INTRODUCTION)

The authors explain the relationship between the forces required to achieve a sufficient press fit and excessive forces that lead to periprosthetic femoral fractures. In the experiments, however, the junction between the femoral head and the stem was investigated. These are two different situations. Please explain why you describe risks and instruments for the insertion of the stem or cavity preparation, but examine the taper junction in your study. (line 37-54)

RESPONSE 1

Thank you for highlighting this. Indeed, you are right that periprosthetic fractures are extremely rare when assembling the head taper junction. The head taper junction was examined in this study as example to investigate the decrease in peak forces caused by the instruments. The sole usage of metal implant components was intentionally chosen, as this helped to exclude any influences of biological specimens while still providing a comparable impaction scenario.

For a better understanding, the following has been added at the end of the introduction (line 53):

For simplicity reasons, the force transmission during the in vitro assembly of the head taper junction between stem taper and the femoral metal ball head was analyzed, focusing explicitly on the force transmission rather than the taper junction itself.

Additionally, the first paragraph of the discussion brings up the topic again.

COMMENT 2 (MATERIALS & METHODS)

The measurement signals were digitally filtered after the measurement to suppress noise. The cut-off frequencies are in the range of the frequencies contained in the force signals (period duration of 0.1 ms leads to frequencies of >10 kHz). Has it been investigated whether the cut-off frequencies influence the shape of the measured signals? This may have an influence on the goodness of fit between measurement and simulation.

RESPONSE 2

Many thanks for the valuable comment. The shortest impulses occurred for the mallet signal (impulse duration: 0.14 ms, period time: 0.28 ms, frequency: 3.6 kHz) and were filtered with a cut-off frequency of 20 kHz. This was necessary to exclude the oscillation of the mallet tip on the stiffness of the force sensor (~25 kHz).

The same applied for the impactor force (impulse duration: 0.27 ms, period duration: 0.54 ms, frequency: 1.9 kHz). Here the oscillation of the impactor handle on the stiffness of the force sensor (10.1 kHz) needed to be filtered without affecting the signal too much. As a result, the cut-off frequency of the filter was defined below the oscillation but as far above the relevant frequencies of the signal as possible.

Nevertheless, you are right and the cut-off frequencies are not orders of magnitude higher than the relevant frequencies of the signals. The following sentence now points this out in the discussion (line 237):

The cut-off frequencies of the mallet and impactor filters were about five times higher than the most dominant frequency of the corresponding signal. This ratio was sufficient to filter the desired frequencies without substantially distorting the signals.

COMMENT 3 (DISCUSSION)

Is it necessary to measure the force in the PoI (head taper junction)? Previous investigations have shown a linear relationship between forces on the stem and turn-off moments or pull-off forces.

RESPONSE 3

We agree with your quoted investigations but the focus of the present manuscript is on the force transmission while the assembly of the head taper junction was only used as an example. The change made with regard to comment 1 hopefully makes this sufficiently clear.

COMMENT4 (CONCLUSION)

You mention that, based on your findings, it is necessary to provide all the information about the instruments used in experimental studies. In your opinion, is it possible to convert study results via the model and thus make them comparable?

RESPONSE 4

Thank you for this interesting question. Yes, within the presented accuracy and based on the assumption, that sufficient information about the instruments is provided, converting results from different setups would be possible.

COMMENT 5 (CONCLUSION)

The implant-bone interface was neither mentioned in the results nor in the discussion. Therefore, it cannot be stated in the conclusion that the forces at the implant-bone interface can be determined more accurately. The corresponding interface has not been modelled in the mechanical model. In my opinion, both the model and the experimental setup would have to be extended in order to be able to make such a statement.

RESPONSE 5

You are totally right. The sentence needs to be formulated more precisely that it would be an option but was not done in the study. We changed it to ”The model could also allow a more accurate determination of the forces at other PoIs, for example the implant-bone interface.” (line 285).

We thank you for your detailed review of our manuscript.

FURTHER CHANGES:

We wish to incorporate the following further changes to our manuscript:

The reference for instrumented impactors (line 48) was changed to Glismann K, Konow T, Huber G, Lampe F, Ondruschka B, Morlock MM. Small design changes affecting primary stability in fully coated tapered wedge stems. PLOS ONE. 2024:Forthcoming.

During the revision step, we noticed three incorrect values in table 2 and corrected them (impactor stiffness, mass of the load cell, impactor tip stiffness).

---

## [Decision Letter · Decision Letter 1]

30 Apr 2024

The force at the implant cannot be assessed by the mallet force – unless supported by a model

PONE-D-23-37167R1

Dear Dr. Schlieker,

We’re pleased to inform you that your manuscript has been judged scientifically suitable for publication and will be formally accepted for publication once it meets all outstanding technical requirements.

Kind regards,

John Leicester Williams, Ph.D.

Academic Editor

PLOS ONE

Additional Editor Comments (optional):

Reviewers' comments:

Reviewer's Responses to Questions

**Comments to the Author**

1. If the authors have adequately addressed your comments raised in a previous round of review and you feel that this manuscript is now acceptable for publication, you may indicate that here to bypass the “Comments to the Author” section, enter your conflict of interest statement in the “Confidential to Editor” section, and submit your "Accept" recommendation.

Reviewer #1: All comments have been addressed

Reviewer #2: All comments have been addressed

2. Is the manuscript technically sound, and do the data support the conclusions?

Reviewer #1: Yes

Reviewer #2: Yes

3. Has the statistical analysis been performed appropriately and rigorously? 

Reviewer #1: N/A

Reviewer #2: Yes

4. Have the authors made all data underlying the findings in their manuscript fully available?

Reviewer #1: Yes

Reviewer #2: Yes

5. Is the manuscript presented in an intelligible fashion and written in standard English?

Reviewer #1: Yes

Reviewer #2: Yes

6. Review Comments to the Author

Reviewer #1: all tha concerns were covered in the reviewed verson of the manuscript, the manuscript could be publish in the present form

Reviewer #2: The authors have addressed all points and answered them adequately. As a note for future work, I suggest that the authors use the same digital filters for the measured and simulated signals to make them fully comparable.

7. PLOS authors have the option to publish the peer review history of their article (what does this mean?). If published, this will include your full peer review and any attached files.

Reviewer #1: **Yes: **Bernardo Innocenti

Reviewer #2: No
